# Glial-Cell-Line-Derived Neurotrophic Factor Promotes Glioblastoma Cell Migration and Invasion via the SMAD2/3-SERPINE1-Signaling Axis

**DOI:** 10.3390/ijms251810229

**Published:** 2024-09-23

**Authors:** Xiaoxiao Guo, Han Zhou, Yifang Liu, Wei Xu, Kouminin Kanwore, Lin Zhang

**Affiliations:** 1Department of Basic Medicine, Kangda College of Nanjing Medical University, Lianyungang 222000, China; xiaoxiao301810@163.com; 2Xuzhou Key Laboratory of Neurobiology, Department of Neurobiology and Anatomy, Xuzhou Medical University, Xuzhou 221004, China; zhouhanzz@163.com (H.Z.); lyvonne1223@163.com (Y.L.); xw755178986@163.com (W.X.); koumininkanwore@gmail.com (K.K.); 3School of Nursing, Xuzhou Medical University, Xuzhou 221004, China

**Keywords:** glioblastomas, GDNF, SERPINE1, SMAD2/3, cell migration, cell invasion

## Abstract

Glial-cell-line-derived neurotrophic factor (GDNF) is highly expressed and is involved in the malignant phenotype in glioblastomas (GBMs). However, uncovering its underlying mechanism for promoting GBM progression is still a challenging work. In this study, we found that serine protease inhibitor family E member 1 (SERPINE1) was a potential downstream gene of GDNF. Further experiments confirmed that SERPINE1 was highly expressed in GBM tissues and cells, and its levels of expression and secretion were enhanced by exogenous GDNF. SERPINE1 knockdown inhibited the migration and invasion of GBM cells promoted by GDNF. Mechanistically, GDNF increased SERPINE1 by promoting the phosphorylation of SMAD2/3. In vivo experiments demonstrated that GDNF facilitated GBM growth and the expressions of proteins related to migration and invasion via SERPINE1. Collectively, our findings revealed that GDNF upregulated SERPINE1 via the SMAD2/3-signaling pathway, thereby accelerating GBM cell migration and invasion. The present work presents a new mechanism of GDNF, supporting GBM development.

## 1. Introduction

Glioblastomas (GBMs) are the most malignant intracranial tumors. Because of GBMs’ strong migratory and invasive abilities, current therapies have limited effectiveness [1]. Increasing research reveals that GBM progression is closely related to multiple cytokines and trophic factors, including glial-cell-line-derived neurotrophic factor (GDNF) [2,3]. GDNF, a member of the transforming growth factor-beta (TGF-β) superfamily, was first isolated from the rat B49 glioma cell line [4]. GDNF has long been considered a nutritional factor that nourishes and protects various neuron populations. However, recent evidence has uncovered that GDNF is abnormally increased in human GBM tissues and cells [5] and enhances GBM cell proliferation, migration, and invasion via multiple mechanisms [6,7,8]. Furthermore, suppressing GDNF and its receptor (GFRα1) effectively slows GBM progression [9], suggesting that GDNF could be a potential target for GBM treatment. Therefore, it is essential to elucidate the underlying mechanism of GDNF-promoted GBM development.

Our previous study has verified that GDNF upregulated serine protease inhibitor family E member 1 (SERPINE1) in rat primary astrocytes [3]. Meanwhile, SERPINE1 was found to be elevated in rat C6 glioma cells treated with GDNF from our early RNA-seq data and enriched in the GO term of wound healing, which indicated that GDNF may accelerate GBM cell migration and invasion through SERPINE1. SERPINE1 is also known as plasminogen activator inhibitor-1 (PAI-1) and was originally considered as a tumor suppressor. Emerging evidence illustrates that SERPINE1 is a promising prognostic marker for many types of tumors [10,11,12], and reducing the SERPINE1 expression significantly inhibits tumor growth [13,14]. Similarly, SERPINE1 is also positively associated with poor prognoses [15] and abnormal cell migration and invasion in GBMs [16]. Nonetheless, whether GDNF facilitates GBMs’ malignant progression via SERPINE1 and its possible mechanism remain elusive.

SMAD2 and SMAD3 (SMAD2/3), two structurally similar proteins, are the key regulators of the TGF-β-signaling pathway responsible for cell proliferation, migration, and differentiation [17]. SMAD2/3 has been reported to be involved in tumor invasion and migration [18,19]. The dynamics of SMAD2/3 phosphorylation is the critical mechanism for modulating the TGF-β-signaling pathway. The interaction between TGF-β and its receptor can drive SMAD2/3 phosphorylation, thereby regulating the levels of target genes [20]. For example, the TGF-β-SMAD2/3-signaling pathway inhibits GBM cell apoptosis by modulating the expression of its downstream gene (Ki-67) [21]. Considering that GDNF is a member of the TGF-β superfamily, we speculated that GDNF may have a similar function for activating SMAD2/3 to TGF-β. Moreover, some studies support that SMAD2/3 are potential upstream regulators of SERPINE1 [15,22]. Hence, it was hypothesized that GDNF may upregulate SERPINE1 by inducing SMAD2/3 phosphorylation to promote GBM cell migration and invasion.

The present study aimed to explore the mechanism by which GDNF aggravates GBM progression. First, the effect of GDNF on SERPINE1 levels was assessed in GBM cells. Subsequent experiments were performed to investigate whether SERPINE1 is involved in GDNF-induced migration and invasion. Next, the molecular mechanism for GDNF regulating the SERPINE1 expression was explored. Finally, in vivo experiments were conducted to verify the above in vitro results. This work clarifies that GDNF enhances GBM cell invasion and migration through the SMAD2/3-SERPINE1 pathway, providing a novel idea for GBM therapy.

## 2. Results

### 2.1. SERPINE1 Is Upregulated in GBM Tissues and Cells

To clarify the mechanism by which GDNF enhances GBM cell migration and invasion, we analyzed the previous RNA-seq data of C6 cells treated with GDNF or PBS [23]. The upregulated differentially expressed genes (DEGs) in the GDNF-treated group were chosen to conduct gene ontology (GO) enrichment analysis. Based on *p*-values, the top 30 GO terms with the most significant enrichments are shown (Figure 1A). Further analysis showed that four genes (SERPINE1, Fmod, Cx3cl1, and Ptk7) were enriched in the GO term of wound healing associated with migration and invasion. SERPINE1 was reported to be highly expressed and participated in tumor progression in GBMs [16]. To verify the relationship between GDNF and SERPINE1, the levels of SERPINE1 in GBMs were first determined using the UALCAN database and then tested in GBM cells and tissues. SERPINE1 mRNA expressions in the tumor and corresponding normal tissues were diverse in various cancers (Appendix A). Nonetheless, the mRNA levels of SERPINE1 in the GBM samples were significantly higher than those in the normal samples (Figure 1B and Appendix A). The survival analysis indicated that the overall survival of GBM patients was low in the high-SERPINE1-expression group compared with the low-SERPINE1-expression group (Figure 1C). In order to confirm the level of SERPINE1 in GBMs, the normal brain (NB) tissues and GBM tissues were collected. In comparison with the normal tissues, SERPINE1 was increased in the GBM tissues (Figure 1D). Consistently, SERPINE1 was higher in rat C6 cells and human GBM cells (U251, U87, and LN229) than in rat astrocytes (RAs) and human astrocytes (HAs), respectively (Figure 1E). These data suggested increased SERPINE1 in GBMs.

### 2.2. GDNF Promotes SERPINE1 Expression and Secretion in GBM Cells

Given that SERPINE1 was elevated in GBMs and may be a downstream gene of GDNF according to RNA-seq results, we ought to investigate the effect of GDNF on regulating SERPINE1. After C6 and U251 cells were treated with different concentrations of GDNF (0, 20, 40, 80, 100 ng/mL) for 48 h [24], the protein expressions and release of SERPINE1 were separately detected by Western blotting and ELISA. For C6 cells, 80 ng/mL GDNF profoundly increased SERPINE1 protein expression, while the protein level of SERPINE1 was remarkably upregulated by 20 ng/mL GDNF in U251 cells (Figure 2A). In line with the above results, an ELISA assay indicated that 48 h of treatment with 80 and 20 ng/mL GDNF facilitated SERPINE1 secretion in C6 and U251 cells, respectively (Figure 2B). In subsequent experiments, the concentration of GDNF was determined to be 80 ng/mL for C6 cells and 20 ng/mL for U251 cells, and the treatment period was 48 h.

### 2.3. SERPINE1 Knockdown Suppresses GDNF-Enhanced GBM Cell Migration and Invasion

To demonstrate whether SERPINE1 is involved in GDNF-mediated cell migration and invasion in GBMs, C6 cells were infected with the lentivirus and U251 cells were transfected with small interfering RNA (siRNA), which inhibited SERPINE1 expression. The results showed that RNAi-3 markedly decreased SERPINE1 mRNA and protein expressions in C6 and U251 cells (Figure 3A,B). Therefore, RNAi-3 was used for subsequent experiments in both C6 and U251 cells. Next, the influences of SERPINE1 knockdown on GDNF-induced cell migration were evaluated by the scratch and transwell assays. SERPINE1 deficiency restricted the migration activities of C6 and U251 cells, while exogenous GDNF strengthened their migration activities (Figure 3C–F). Furthermore, SERPINE1 knockdown significantly reversed increased cell migration caused by GDNF in C6 and U251 cells (Figure 3C–F). Following transwell matrigel assay revealed that the invasion activities of C6 and U251 cells were decreased by knocking down SERPINE1 and enhanced by the addition of GDNF (Figure 3G). Knocking down SERPINE1 significantly inhibited the pro-invasion effects of GDNF on C6 and U251 cells (Figure 3G). These results indicated that GDNF may accelerate the migration and invasion of GBM cells via SERPINE1.

### 2.4. GDNF Increases SERPINE1 Expression via SMAD2/3

It was reported that SMAD2/3 were potential upstream regulators of SERPINE1 [15]. To explore whether GDNF promotes SERPINE1 expression via the SMAD2/3-signaling pathway, the phosphorylation levels of SMAD2 and SMAD3 were examined by Western blotting following GDNF treatments for 0, 1, 2, and 4 h [25]. After treatment with GDNF for 4 h, phosphorylated SMAD2 and SMAD3 were significantly elevated in C6 and U251 cells (Figure 4A,B). Subsequently, the effects of GDNF in SERPINE1 protein expressions were measured in C6 cells when SMAD2 and SMAD3 were inhibited by siRNA. The results showed that inhibition of SMAD2 and SMAD3 significantly downregulated SERPINE1 expression (Figure 4C). Of note, increased SERPINE1 resulting from the addition of GDNF was reduced when SMAD2 and SMAD3 were silenced in C6 cells (Figure 4C), uncovering that GDNF may upregulate SERPINE1 through the SMAD2/3-signaling pathway.

### 2.5. GDNF Facilitates GBM Growth In Vivo by SERPINE1

An in vivo experiment was performed by subcutaneous injection of C6 cells stably transfected with the lentivirus of SERPINE1 knockdown and GDNF to nude mice. Two weeks after injection, mice were euthanized, and the tumor tissues were collected (Figure 5A). Further analysis suggested that the tumor volume and weight were significantly increased and decreased by GDNF and SERPINE1 deficiency, respectively (Figure 5B,C). SERPINE1 knockdown suppressed tumor progression promoted by GDNF (Figure 5B–D). Additionally, immunohistochemistry (IHC) assay showed that silencing SERPINE1 inhibited the expressions of Ki-67, GFAP, MMP2 and MMP9, while GDNF promoted their expressions (Figure 5E,F and Appendix A). Furthermore, the knockdown of SERPINE1 reduced the levels of Ki-67, GFAP, MMP2 and MMP9 elevated by GDNF (Figure 5E,F and Appendix A). The above findings illustrated that GDNF aggravated GBM growth in vivo via SERPINE1.

## 3. Discussion

GDNF is a vital factor that promotes GBM occurrence and development [23,26], but its underlying mechanisms have not been fully investigated. In the present study, we validated a feasible mechanism of GDNF-induced abnormal cell migration and invasion and uncovered the important role of GDNF-SMAD2/3-SERPINE1 axis in GBM progression. SERPINE1 was highly expressed in GBM tissues and cells, and mediated GDNF-enhanced GBM cell migration and invasion, and in vivo growth. Mechanistically, GDNF promoted SMAD2/3 phosphorylation and activated the SMAD2/3-signaling pathway, and in turn, SMAD2/3 induced a SERPINE1 increase.

To explore how GDNF accelerates cell invasion and migration, our previous RNA-seq data from C6 cells treated with or without GDNF were analyzed. The results showed that the GO term of wound healing was significantly enriched in the GDNF-treated group, and correlated with cell migration and invasion. Among this GO term, four genes including SERPINE1 were enriched, which was consistent with our recent publication [3]. Other studies have shown that TGF-β is involved in the induction of SERPINE1 expression in tumor cells [27,28]. Meanwhile, GDNF is known as a member of the TGF-β superfamily. Hence, SERPINE1 may be a potential downstream molecule of GDNF. To verify this issue, we first analyzed the mRNA expressions of SERPINE1 among various tumors using the UALCAN database. Surprisingly, the mRNA expression patterns of SERPINE1 in different tumors were diverse, and its mRNA levels were lower in nearly half of the tumor tissues than the corresponding normal tissues, which indicated that SERPINE1 was specific for tumor types. However, data from the database and subsequent experiments revealed that SERPINE1 was significantly elevated in GBM tissues and cells. A similar finding was reported in previous studies [15,29]. Numerous studies have elucidated that SERPINE1, as a critical regulator of extracellular matrix remodeling, plays an important role in the malignant phenotypes of GBM cells, such as invasion, migration and epithelial-mesenchymal transition (EMT) [16,30]. Nevertheless, it is unclear whether GDNF facilitates tumor progression via SERPINE1 in GBMs.

To answer this, GBM cells from humans (U251) and rats (C6) were treated with various doses of GDNF to investigate the effects of GDNF on SERPINE1 expressions and release. We found that GDNF strikingly increased the expressions and secretion of SERPINE1 both in U251 and C6 cells, suggesting that GDNF, as a member of the TGF-β superfamily, may function like TGF-β. Of note, although the optimal concentration of GDNF action was different in the two types of cells, the results of Western blotting and ELISA assays were consistent in the same type of cells. This finding unraveled the importance of determining the effective concentration of the target molecules before performing more experiments [31]. To ensure the reliability of experimental data, 80 and 20 ng/mL GDNF were used for the treatment of C6 and U251 cells in the following work, respectively. To clarify the roles of SERPINE1, we explored the effects of GDNF on cell migration and invasion of SERPINE1-suppressed U251 and C6 cells. This study illustrated that inhibiting SERPINE1 significantly weakened GBM cell migration and invasion. Moreover, SERPINE1 knockdown effectively reduced GDNF-increased invasion and migration of GBM cells, uncovering that GDNF may promote GBM progression through SERPINE1. To verify the results of in vitro experiments, a mouse subcutaneous xenograft glioma model was established to assess the effects of GDNF and SERPINE1 on GBM growth in this study. However, the orthotopic GBM models could assess the tumor behaviors more accurately. In future research, we will utilize more animal models to explore the effects of GDNF on GBM progression. The next question was how GDNF regulates SERPINE1 in GBM cells.

The TGF-β/SMAD-signaling pathway has been the subject of extensive research due to its vital role in carcinogenesis [32,33]. TGF-β superfamily is the largest family of secreted growth factors, including TGF-β, bone morphogenetic protein (BMP), activin/inhibins, growth differentiation factor (GDF), and GDNF. Indeed, TGF-β signals were activated by the canonical SMAD-dependent pathway and non-canonical SMAD-independent pathway [34]. It was documented that TGF-β regulated SERPINE1 expression via the SMAD pathway [35]. Inversely, another study showed that TGF-β-induced SERPINE1 was independent of the SMAD pathway [36]. However, whether GDNF-mediated SERPINE1 expression is dependent on the SMAD pathway has not been reported. Given that SMAD2/3 phosphorylation is known as a vital mechanism of activating TGF-β signaling, we first tested the effects of GDNF on SMAD2/3 phosphorylation in GBM cells. The results found that GDNF notably promoted the phosphorylation of SMAD2 and SMAD3. To further investigate if GDNF modulates SERPINE1 via SMAD2/3, the effects of GDNF on SERPINE1 expressions were assessed in SMAD2/3-inhibited C6 cells. In the present work, silencing SMAD2/3 reversed elevated SERPINE1 caused by GDNF, indicating that GDNF upregulated SERPINE1 in the SMAD2/3-dependent manner. Similarly, GDF8, another member of the TGF-β superfamily, enhanced the expression of SERPINE1 via SMAD2/3 signaling [37]. Additionally, this study also demonstrated that GDF8 resulted in the activation of SMAD2/3 through activin receptor-like kinase 5 (ALK5). Accumulating evidence has illuminated that SMAD2/3 is regulated by the receptors, and TGF-β binding to its corresponding receptor can lead to SMAD2/3 phosphorylation [38]. However, which receptor is responsible for the GDNF-activated SMAD2/3 pathway remains unclear. Although GFRα1 and RET are the classic receptors of GDNF [39,40], it was reported that GDNF induced the SMAD2/3-signaling pathway by binding the receptor (ALK5) rather than GFRα1 [25]. Therefore, we speculate that GDNF may activate the SMAD2/3-signaling pathway to promote SERPINE1 expression via the ALK5 receptor in GBMs.

In conclusion, our study illustrates that GDNF reinforces GBM cell migration and invasion via the SMAD2/3-SERPINE1 axis. These results deepen the understanding of GDNF promoting GBM progression and provide new ideas for GBM treatment.

## 4. Materials and Methods

### 4.1. Cells and Tissues

The GBM cell lines (C6, U251, U87, and LN229) purchased from the Cell Bank of the Chinese Academy of Sciences were cultured in Dulbecco’s modified eagle medium (DMEM) (Hyclone, Logan, UT, USA) with 10% fetal bovine serum (FBS) (Clark, VA, USA) and 1% penicillin. Astrocytes (RAs and HAs) were provided by the Neurobiology Research Center of Xuzhou Medical University. HAs were cultured in Astrocyte Medium (AM) (Sciencell Research Laboratories, Carlsbad, CA, USA) with 10% FBS and 1% smooth muscle cell growth supplement (SMCGS). RAs were cultured in DMEM (Hyclone) with 10% FBS (Clark). All the cell lines were cultured in 5% CO_2_ at 37 °C.

In this study, C6 and U251 cells were frequently subjected to GDNF treatment. The GDNF recombinant protein (human: P39905-1, MedChemExpress, Monmouth Junction, NJ, USA; rat: QP5504, enQuire Bio, Denver, CO, USA) was dissolved with PBS to a concentration of 10 µg/mL, and stored at −80 °C. In the following experiments, GDNF was added to the cells after being diluted to the desired concentrations using the medium.

GBM and NB tissues were acquired from the Affiliated Hospital of Xuzhou Medical University. Three GBM tissues were collected from the surgical specimen archives, and three NB tissues were obtained from the patients who suffered acute brain damage and underwent intracranial decompression. All protocols were approved by the ethics committee at Xuzhou Medical University.

### 4.2. Western Blotting

Total protein from cells and tissues was extracted by RIPA lysis buffer (Beyotime, Shanghai, China) containing protease and phosphatase inhibitors (Beyotime). Protein samples were separated by 10% SDS-PAGE and then transferred to the NC membranes (Vicmed, Xuzhou, China). The membranes were blocked in 5% skimmed milk for 2 h, and incubated in the primary antibodies against SERPINE1 (ab222754, Abcam, Cambridge, UK), SMAD2 (bs-0718R, Bioss, Beijing, China), phospho-SMAD2 (Ser465/467) (#18338, CST, Danvers, MA, USA), SMAD3 (bs-348R, Bioss), phospho-SMAD3 (Ser423/425) (#9520, CST), and GAPDH (60004-1-Ig, Proteintech, Rosemont, IL, USA) overnight at 4 °C. Next, the blots were incubated in the secondary antibodies at room temperature for 1 h and scanned using the Odyssey Classic Infrared Imaging System (Li-Cor, Lincoln, NE, USA).

### 4.3. Quantitative Real-Time PCR (q-PCR)

Total RNA from cells was isolated using TRIzol reagent (Invitrogen, Carlsbad, CA, USA). The first-strand cDNA was synthesized using a cDNA synthesis kit (Vazyme, Nanjing, China). The cDNA was analyzed by q-PCR (SYBR Green, Invitrogen) with an Applied SYBR Sequence Detection System (Light Cycler 480 II, Roche). GAPDH was used as an internal control. The relative expressions of the targeted genes were analyzed with the 2^−ΔΔCT^ method (Appendix A). All primers were obtained from Sangon (Shanghai, China) and their sequences are available in Appendix A.

### 4.4. Enzyme-Linked Immunosorbent Assay (ELISA)

The concentrations of extracellular SERPINE1 in C6 and U251 cells were evaluated using the rat and human SERPINE1 ELISA Kits (USCNK, Wuhan, China) as previously described [16]. Each treatment and assay were performed in triplicate and repeated three times.

### 4.5. Wound Healing Assay

Cell migration was determined by the wound healing assay. Briefly, C6 and U251 cells were cultured in a 6-well plate. When the cell confluency was more than 95%, cells were scratched with a 200 ul pipette tip. Next, cells were rinsed with PBS and maintained in a serum-free medium. Cells were photographed by the microscope at 0 and 48 h after scratching, and the wound area was calculated by Image J.

### 4.6. Transwell Migration and Invasion Assay

The transwell invasion and migration assay were applied to assess cell invasion and migration, respectively. The only difference was that the upper transwell chamber was coated by the matrigel for the invasion assay, but not coated by the matrigel for the migration assay. Then, cells were cultured in the upper chamber with a serum-free medium, while a medium containing 10% FBS was added into the lower chamber. After 48 h of incubation, non-invading or non-migrating cells were gently wiped, and invading or migrating cells were fixed and stained. Finally, stained cells were counted under the microscope.

### 4.7. RNA Lentivirus, siRNA and Transfection

siRNAs, including si-SERPINE1 (human), si-SMAD2 (rat), si-SMAD3 (rat), and corresponding si-Control, were obtained from GenePharma Technology (Suzhou, China). GBM cells were transfected with indicated siRNAs using Lipofectamine 3000 (Invitrogen) based on the instructions. siRNAs targeting the rat SERPINE1 mRNA as well as its si-Control were cloned into lentivirus-based vectors (GeneChem, Shanghai, China). Stable C6 cell lines obtained by flow cytometer (Appendix A) were used for the subsequent construction of the mouse model. si-Control was applied as the negative control (NC). The siRNAs’ sequences are available in Appendix A.

### 4.8. Mouse Subcutaneous Xenograft Glioma Model

BALB/c female nude mice (*n* = 20, aged 4–6 weeks) were purchased from Nanjing Jicui Company. These mice were randomly divided into 4 groups: NC+PBS, NC+GDNF, SERPINE1i+PBS, and SERPINE1i+GDNF. C6 cells were subcutaneously injected into the nude mice following the standard procedures. Meanwhile, PBS or GDNF (0.1 μg for each mouse) was injected into the nude mice every other day. All mice were monitored for 2 weeks. The mouse weight and tumor volume were measured every other day. Two weeks after injection, the tumor masses were isolated and weighed. The animal study was carried out in accordance with the Guide for the Care and Use of Laboratory Animals by NIH (No. 85-23, 1996). All work was approved by the Animal Ethics Committee of Xuzhou Medical University (202202A338).

### 4.9. Hematoxylin–Eosin (HE)

The tumor tissues isolated from the mice following euthanasia were used for HE staining. Briefly, after the tumor tissues fixed in 10% formalin were dehydrated and embedded, they were sliced to obtain the paraffin sections (5 µm). The sections were stained for 2 min in hematoxylin solution following deparaffinization and rehydration. Subsequently, the sections were differentiated for 5 s, returned to blue for 20 s, and then stained in eosin solution for 1 min. Finally, the sections were stepwise dehydrated in increasing concentrations of alcohol and imaged under an upright optical microscope (Nikon, Tokyo, Japan).

### 4.10. Immunohistochemistry (IHC)

The paraffin sections obtained according to the methods described in HE staining were also used for IHC staining. The sections were deparaffinized, rehydrated, and then heated in citrate buffer at 121 °C for 30 min. After incubating with 0.3% hydrogen peroxide (PV-9001, OriGene, Rockville, MD, USA) for 30 min, the sections were incubated in the primary antibodies at 4 °C overnight. All primary antibodies were purchased from Proteintech (Rosemont, IL, USA), including Ki-67 (28074-1-AP), GFAP (81063-1-PBS), MMP2 (66366-1-PBS), and MMP9 (27306-1-AP). Subsequently, the sections were washed with PBS, and incubated in DAB solution (ZLI-9018, OriGene) at 37 °C for 2 h. Finally, the sections were imaged by an upright optical microscope (Nikon).

### 4.11. Bioinformatics Analysis

To find the potential downstream genes of GDNF, our early RNA-seq data were applied for the GO enrichment analysis of upregulated DEGs using the DAVID database (https://david.ncifcrf.gov/tools.jsp) accessed on 29 March 2021. In addition, the UALCAN database (https://ualcan.path.uab.edu/analysis.html) accessed on 31 August 2023 was used to analyze the mRNA levels of SERPINE1 in human normal and tumor tissues, including GBMs. To determine the correlation between the GBM patients’ outcome and SERPINE1 mRNA expression, the survival analysis was performed using the Oncomine database (https://www.oncomine.org/resource/login.html) accessed on 29 March 2021.

### 4.12. Statistical Analysis

Data are expressed as mean ± standard error (SE) and analyzed using SPSS 25.0 (IBM Corp. Armonk, NY, USA) and Graphpad Prism 8.0 (San Diego, CA, USA). The mean values between the two groups were compared by independent sample *t*-tests. One-way analysis of variance (ANOVA) was employed to determine the significant differences among multiple groups. In all tests, a *p*-value < 0.05 was considered statistically significant.

## Figures and Tables

**Figure 1 ijms-25-10229-f001:**
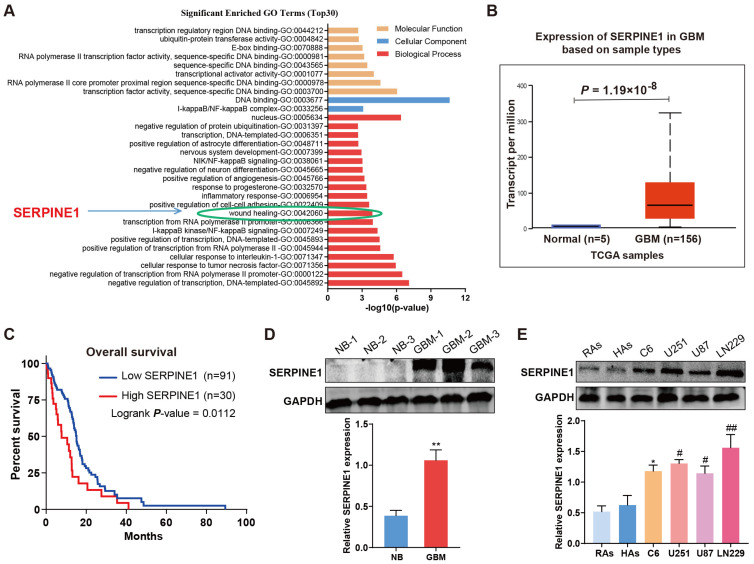
SERPINE1 is increased in GBMs. (**A**) GO enrichment analysis of upregulated DEGs in C6 cells treated with GDNF. The top 30 significant GO terms were exhibited according to *p*-value. (**B**) SERPINE1 mRNA expressions were compared between the GBM and normal tissues from the UALCAN database. (**C**) The overall survival was analyzed to evaluate the association between the mRNA level of SERPINE1 and the outcome of GBM patients by the Oncomine website. (**D**,**E**) The protein expressions of SERPINE1 in GBM tissues and cells were determined by Western blotting. NB: normal brain; RAs: rat astrocytes; HAs: human astrocytes (vs. NB: **, *p* < 0.01. vs. RAs: *, *p* < 0.05. vs. HAs: #, *p* < 0.05; ##, *p* < 0.01).

**Figure 2 ijms-25-10229-f002:**
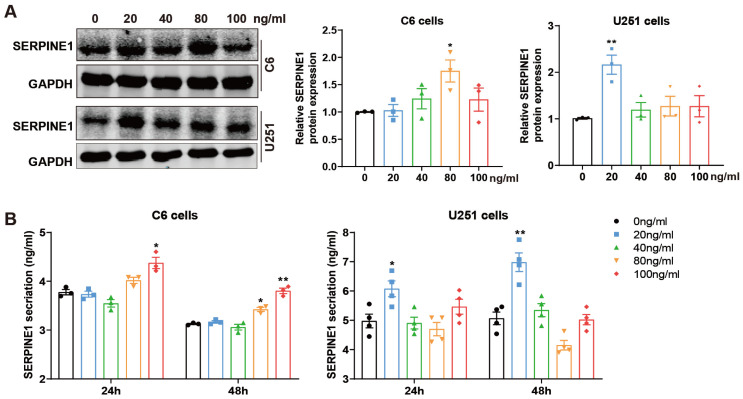
GDNF promotes SERPINE1 expressions and secretion in GBM cells. (**A**) Western blotting (left) was used to detect the protein expressions of SERPINE1 in C6 and U251 cells treated with various doses of GDNF (0, 20, 40, 80, 100 ng/mL) for 48 h. The statistical analysis (right) of the left bands was performed. (**B**) The contents of SERPINE1 were tested in C6 and U251 cells after 24 and 48 h of treatment with different concentrations of GDNF using ELISA kits (vs. 0 ng/mL: *, *p* < 0.05; **, *p* < 0.01).

**Figure 3 ijms-25-10229-f003:**
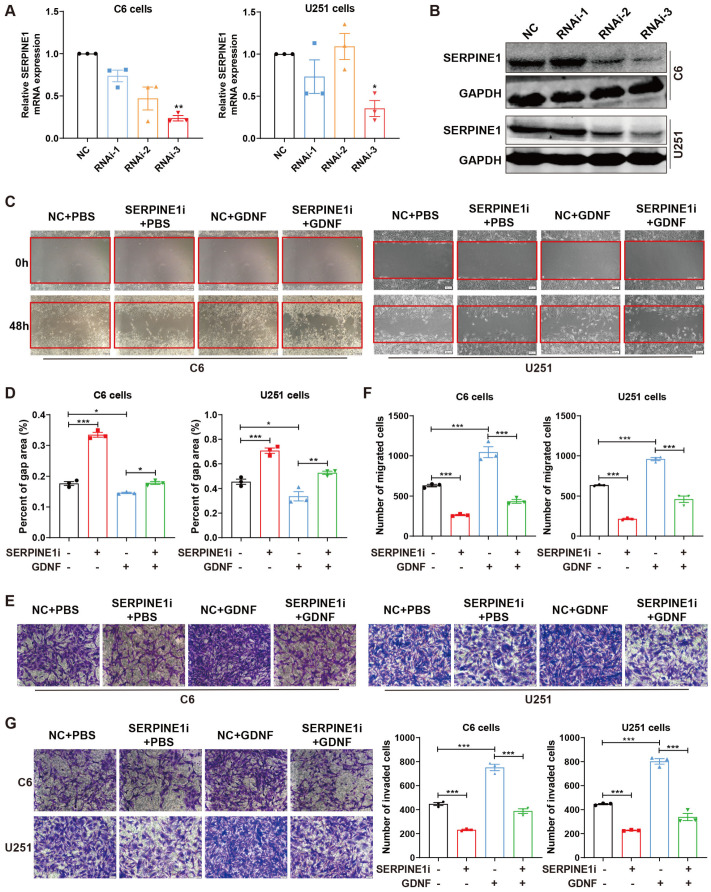
SERPINE1 knockdown strikingly suppresses the migration and invasion of GBM cells enhanced by GDNF. (**A**,**B**) qPCR and Western blotting were applied to assess the knockdown efficiency of SERPINE1 in C6 and U251 cells. (**C**,**D**) Wound healing assay was performed to examine the effects of SERPINE1 deficiency on cell migration in C6 and U251 cells separately treated with 80 and 20 ng/mL GDNF for 48 h. Bar = 100 µm. (**E**,**F**) The migratory abilities were detected by transwell migration assay in SERPINE1 silenced C6 and U251 cells treated with 80 and 20 ng/mL GDNF for 48 h, respectively. Bar = 50 µm. (**G**) Transwell invasion assay was conducted to evaluate the influences of SERPINE1 knockdown on cell invasion in C6 and U251 cells after 48 h of separate treatment with 80 and 20 ng/mL GDNF. Bar = 50 µm (*, *p* < 0.05; **, *p* < 0.01; ***, *p* < 0.001).

**Figure 4 ijms-25-10229-f004:**
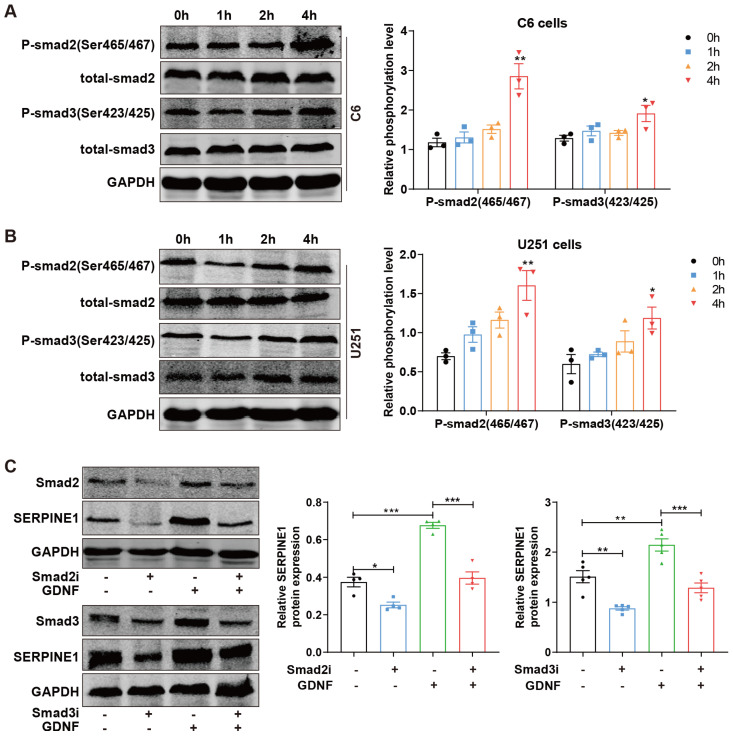
GDNF upregulates SERPINE1 via SMAD2/3 phosphorylation in GBM cells. (**A**,**B**) The levels of SMAD2/3 phosphorylation were analyzed in C6 and U251 cells separately treated with 80 and 20 ng/mL GDNF for 0, 1, 2, and 4 h using Western blotting. (**C**) The effects of SMAD2/3 deficiency on SERPINE1 protein expressions were assessed in C6 cells in the presence of 80 ng/mL GDNF for 48 h by Western blotting (*, *p* < 0.05; **, *p* < 0.01; ***, *p* < 0.001).

**Figure 5 ijms-25-10229-f005:**
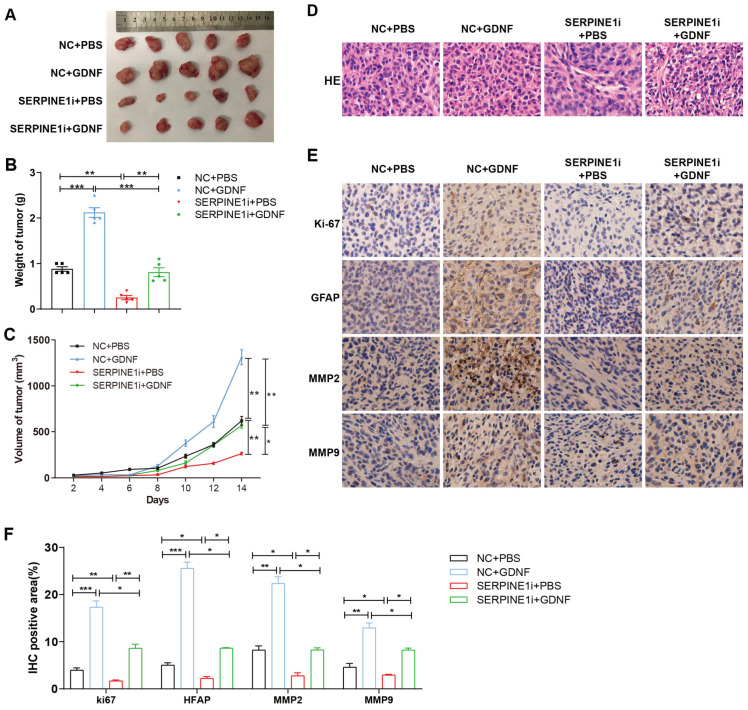
GDNF accelerates GBM growth in vivo. (**A**,**B**) The tumor tissues were exfoliated and weighed at 14 days after subcutaneous tumor formation. (**C**) The mean tumor volumes were calculated every 2 days until 14 days. (**D**) Hematoxylin-Eosin (HE) staining of the tumor tissues was performed (×20). (**E**,**F**) IHC staining was used to detect the levels of Ki-67, GFAP, MMP2 and MMP9 (×20). *n* = 5 (*, *p* < 0.05; **, *p* < 0.01. *** *p* < 0.001).

## Data Availability

The data will be made available upon request.

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
