# Peer review of "Glial-Cell-Line-Derived Neurotrophic Factor Promotes Glioblastoma Cell Migration and Invasion via the SMAD2/3-SERPINE1-Signaling Axis"

_ijms, 2024, doi:10.3390/ijms251810229_

Round 1
Reviewer 1 Report
Comments and Suggestions for Authors
The article explores the molecular mechanisms underlying glioblastoma (GBM) progression. It specifically examines the role of glial cell line-derived neurotrophic factor (GDNF) in promoting the migration and invasion of GBM cells. The authors hypothesize that GDNF upregulates SERPINE1 through the SMAD2/3 signaling pathway, thereby contributing to the malignant phenotype of GBM.
The study addresses a critical gap in the understanding of how GDNF influences GBM progression. By identifying the SMAD2/3-SERPINE1 axis, the authors provide novel insights and suggest potential therapeutic targets. The research is grounded in a combination of in vitro and in vivo experiments. However, the manuscript has several shortcomings that hinder its clarity and scientific rigor.
Major Concerns:
The authors do not provide explicit details about the source and preparation of GDNF. It is essential to specify whether GDNF was purchased and, if so, how it was dissolved, including the solvent used. A detailed section on this should be added to the Materials and Methods. Likewise, the authors do not provide a material and methods section about HE staining and Immunohistochemistry of tumor samples.
Figure 1A. The datasets used for the GO enrichment analysis of up-regulated DEGs in C6 cells treated with GDNF are not specified. The authors mention on line 328 that their "early RNA-seq data were applied for gene ontology enrichment analysis of up-regulated differentially expressed genes using the DAVID database." However, they do not provide the reference for this early RNA-seq data. Is this previously published data? The authors should include the reference and specify the GEO accession code of the datasets used.
Figure 1B. The bar plot presented needs clarification regarding the statistical significance of the differences shown. The authors should indicate whether these differences are statistically significant. Additionally, the date of access for the data obtained from the UALCAN web tool should be included.
Figure 1C. Figure 1C appears redundant with Figure 1B. The authors should consider combining these figures or provide additional distinct information to justify the inclusion of both.
Figure 2A. The GAPDH blot is overexposed, making it difficult to assess the protein amounts loaded in the lanes. The authors should provide images with lower exposure times to accurately represent the protein levels. Additionally, the plots on the right side of Figure 2A need to be explained in the text and described in the figure caption. Specifically, the authors should clarify why the control groups in the plots start with expression levels different from 1.
Figure 3A.The authors state that the qPCR data were evaluated using the 2−ΔΔCT method. To validate this analysis, the primer efficiency curves for the primer sets used should be provided as a supplementary figure.
The authors state on line 154 that GDNF treatment was conducted for 4 hours, but in the figure caption of Figure 4A (line 164), they mention treatment durations of 80 and 20 ng/mL GDNF for 48 hours. This inconsistency should be clarified.
Lines 153-154: The authors mention that phosphorylated SMAD2 and SMAD3 levels were elevated after GDNF treatment for 4 hours, but they do not show a vehicle control for the 4-hour treatment. This control is necessary to determine whether the increase in phosphorylated SMAD is a normal cellular response.
Figure 5. The HE staining and immunohistochemistry images do not clearly reflect the results described in the text. The authors should provide higher magnification images (at least 20X) to enhance clarity.
Line 313: The authors mention that "Stable C6 cell lines obtained by flow cytometer were used for the subsequent construction of the mouse model." However, they do not present the results of this sorting by FACS. These results should be included as a supplementary figure at least.
The reliance on overexpression and knockdown experiments may introduce artifacts. For instance, the high concentrations of GDNF used in vitro might not accurately reflect physiological conditions. It would be beneficial to rigorously examine the dose-response relationship and use additional controls to rule out off-target effects.
The in vivo experiments use a subcutaneous xenograft model, which does not fully replicate the brain tumor microenvironment. The use of orthotopic GBM models would provide a more accurate assessment of tumor behavior and its interaction with surrounding brain tissues.
A concluding paragraph should be added at the end of the discussion section to summarize the key findings and their implications.
Minor Revisions:
Figure 1D. It is unclear whether the plot shows overall survival or disease-free survival. This should be clearly stated in the figure caption and corrected accordingly.
Figure 1E. The figure caption does not clarify what "NB-1," "NB-2," and "NB-3" represent. Please specify this.
The authors should minimize the use of adjectives such as "strikingly" and "evidently" throughout the text.
The manuscript should be carefully reviewed for typographical errors.
Abbreviations should be defined or written in full upon their first appearance only. For instance, "FBS" is first mentioned on line 260 and repeated in line 265.
Author Response
Comments and Suggestions for Authors
The article explores the molecular mechanisms underlying glioblastoma (GBM) progression. It specifically examines the role of glial cell line-derived neurotrophic factor (GDNF) in promoting the migration and invasion of GBM cells. The authors hypothesize that GDNF upregulates SERPINE1 through the SMAD2/3 signaling pathway, thereby contributing to the malignant phenotype of GBM.
The study addresses a critical gap in the understanding of how GDNF influences GBM progression. By identifying the SMAD2/3-SERPINE1 axis, the authors provide novel insights and suggest potential therapeutic targets. The research is grounded in a combination of in vitro and in vivo experiments. However, the manuscript has several shortcomings that hinder its clarity and scientific rigor.
- Summary
Thank you very much for your valuable comments concerning our manuscript entitled “GDNF promotes glioblastoma cell migration and invasion via SMAD2/3-SERPINE1 signaling axis” (ID: ijms-3169672). Those comments are all valuable and helpful for revising and improving our paper, as well as the important guiding significance to our researches. We have studied comments carefully and made corrections which we hope meet with approval. Also, we have went through the manuscript carefully and improved it. Revised portion are marked in red in the paper. The main corrections in the paper and responses to the reviewer’s comments are as followed.
2. Questions for General Evaluation |
Reviewer’s Evaluation |
Response and Revisions |
Does the introduction provide sufficient background and include all relevant references? |
Can be improved |
Agree. We have revised it. |
Is the research design appropriate? |
Can be improved |
Agree. We have revised it. |
Are the methods adequately described? |
Must be improved |
Agree. We have revised it. |
Are the results clearly presented? |
Must be improved |
Agree. We have revised it. |
Are the conclusions supported by the results? |
Must be improved |
Agree. We have revised it. |
- Point-by-point response to Comments and Suggestions for Authors
Major Concerns:
Comments 1: The authors do not provide explicit details about the source and preparation of GDNF. It is essential to specify whether GDNF was purchased and, if so, how it was dissolved, including the solvent used. A detailed section on this should be added to the Materials and Methods. Likewise, the authors do not provide a material and methods section about HE staining and Immunohistochemistry of tumor samples.
Response 1: Thank you for your valuable suggestions and careful work. Sorry. It is an oversight in our manuscript. We have added the details of GDNF application, HE staining and IHC staining to “Materials and methods” section in the revised manuscript.
-GDNF application (page 9, line 277-281)
In this study, C6 and U251 cells were frequently subjected to GDNF treatment. The GDNF recombinant protein (human: P39905-1, MedChemExpress, USA; rat: QP5504, enQuire BioReagents, USA) was dissolved with PBS to a concentration of 10 µg/ml, and stored at -80℃. In the following experiments, GDNF was added into cells after being diluted to the desired concentrations using the medium.
-Hematoxylin–eosin (HE) (page 11, line 347-355)
Tumor tissues isolated from the mice following euthanasia were used for HE staining. Briefly, after tumor tissues fixed in 10% formalin were dehydrated and embedded, they were sliced to obtain the paraffin sections (5 µm) of GBM tissues. The sections were stained for 2 min in hematoxylin solution following deparaffinization and rehydration. Subsequently, the sections were differentiated for 5 s, returned to blue for 20 s and then stained in eosin solution for 1 min. Finally, the sections were stepwise dehydrated in increasing concentrations of alcohol, and imaged under an upright optical microscope (Nikon, Japan).
-Immunohistochemistry (IHC) (page 11, line 356-365)
The paraffin sections obtained according to the methods described in HE staining were also used for IHC staining. The sections were deparaffinized, rehydrated, and then heated in citrate buffer at 121°C for 30 min. After incubating with 0.3% hydrogen peroxide (PV-9001, Origene, Maryland, USA) for 30 min, the sections were incubated in primary antibody at 4°C overnight. All primary antibodies were purchased from Proteintech (Rosemont, IL, USA), including Ki-67 (28074-1-AP), GFAP (81063-1-PBS), MMP2 (66366-1-PBS) and MMP9 (27306-1-AP). Subsequently, the sections were washed with PBS, and incubated in DAB solution (ZLI-9018, OriGene) at 37°C for 2 h. Finally, the sections were imaged by an upright optical microscope (Nikon).
Comments 2: Figure 1A. The datasets used for the GO enrichment analysis of up-regulated DEGs in C6 cells treated with GDNF are not specified. The authors mention on line 328 that their "early RNA-seq data were applied for gene ontology enrichment analysis of up-regulated differentially expressed genes using the DAVID database." However, they do not provide the reference for this early RNA-seq data. Is this previously published data? The authors should include the reference and specify the GEO accession code of the datasets used.
Response 2: Thank you for pointing this out. The data used in this work have been previously published by our group (PMID: 37594930, title: GDNF triggers proliferation of rat C6 glioma cells via the NF-κB/CXCL1 signaling pathway). This paper has been added to the reference in the revised manuscript (page 2, line 77). There was no GEO accession code because the RNA-seq data was not uploaded to online database. However, the RNA-seq data will be provided if the readers need them. We have added the declaration of data availability in the revised manuscript. The details are as followed.
-Data availability (page 12, line 395)
The data will be made available upon request.
Comments 3: Figure 1B. The bar plot presented needs clarification regarding the statistical significance of the differences shown. The authors should indicate whether these differences are statistically significant. Additionally, the date of access for the data obtained from the UALCAN web tool should be included.
Response 3: Thanks for your valuable comments. Indeed, a clear presentation of statistical differences will help readers understand the paper better. The website does not provide statistical differences in the expression of the individual gene in pan-cancer (PMID: 34527588), although the statistical difference of gene expressions in one certain tumor and normal tissues can be available using UALCAN. If the readers are interested in this, they can use the website in the paper to analyze SERPINE1 expression differences in one certain tumor. Additionally, we have added the date of access to figure caption. The details are as followed.
-SERPINE1 mRNA expression was compared between GBM and normal tissues from UALCAN database on 31 August 2023. (page 3, line 100)
Comments 4: Figure 1C. Figure 1C appears redundant with Figure 1B. The authors should consider combining these figures or provide additional distinct information to justify the inclusion of both.
Response 4: Thank you for your advice. Figure 1B appears SERPINE1 mRNA expression between multiple human cancers and corresponding normal tissues, which analyzed the expression of SERPINE1 in different types of cancers including GBM. The results showed that the mRNA level of SERPINE1 was diverse in various cancers. Figure 1C only appears SERPINE1 mRNA expression in GBM and corresponding normal tissues. In fact, both Figure 1B and 1C are aimed at determining increased SERPINE1 in GBM tissues compared with normal tissues. Given comments 3 and 4, Figure 1B is presented as a supplementary figure 1. Therefore, a new Figure 1 were provided.
-Figure 1 (page 3, line 96)
Figure 1. SERPINE1 is increased in GBM. (A) GO enrichment analysis of up-regulated DEGs in C6 cells treated with GDNF. The top 30 significant GO terms were exhibited according to P-value. (B) SERPINE1 mRNA expression was compared between GBM and normal tissues from UALCAN database on 31 August 2023. (C) Overall survival was analyzed to evaluate the association between the mRNA level of SERPINE1 and the outcome of GBM patients by oncomine website. (D, E) The protein expressions of SERPINE1 in GBM tissues and cells were determined by western blotting. NB: normal brain; RA: rat astrocytes; HA: human astrocytes. (vs NB: **, P < 0.01. vs RA: *, P < 0.05. vs HA: #, P < 0.05; ##, P < 0.01).
-Figure S1 (supplementary materials)
Figure S1. The mRNA levels of SERPINE1 were analyzed between multiple human tumors and corresponding normal tissues using UALCAN database.
Comments 5: Figure 2A. The GAPDH blot is overexposed, making it difficult to assess the protein amounts loaded in the lanes. The authors should provide images with lower exposure times to accurately represent the protein levels. Additionally, the plots on the right side of Figure 2A need to be explained in the text and described in the figure caption. Specifically, the authors should clarify why the control groups in the plots start with expression levels different from 1.
Response 5: Thanks for your valuable comments.
-The GAPDH band has been changed to a short time exposure one. Therefore, a new Figure 2 were provided. (page 4, line 118)
-The plots on the right side of Figure 2A are the statistical results of the left bands. We have explained in the text and described it in the figure caption. The details are as followed. (page 3, line 119-122)
Western blotting (left) was used to detect the protein expressions of SERPINE1 in C6 and U251 cells treated with various doses of GDNF (0, 20, 40, 80, 100 ng/ml) for 48 h. The statistical analysis (right) of the left bands was performed.
-Thanks for your careful work. The expression levels of control groups were not homogenized to 1 in the previous statistical analysis. We have reanalyzed the data and plotted it. Therefore, a new Figure 2 were provided. (page 4, line 118)
Comments 6: Figure 3A.The authors state that the qPCR data were evaluated using the 2−ΔΔCT method. To validate this analysis, the primer efficiency curves for the primer sets used should be provided as a supplementary figure.
Response 6: Thanks for your good suggestions. The amplificaion curves for the primer sets are presented as a supplementary figure 3 in the revised manuscript. (supplementary materials)
Figure S3. The amplificaion curves of SERPINE1 primers in U251 (A) and C6 (B) cells.
Comments 7: The authors state on line 154 that GDNF treatment was conducted for 4 hours, but in the figure caption of Figure 4A (line 164), they mention treatment durations of 80 and 20 ng/mL GDNF for 48 hours. This inconsistency should be clarified.
Response 7: Thanks for your careful work. Sorry. It is an oversight in our manuscript. In the present study, C6 and U251 cells separately treated with 80 and 20 ng/ml GDNF for 0, 1, 2, and 4 h. (page 6, line 157) The western blotting results showed that treatment with GDNF for 4 h significantly promoted the phosphorylation of SMAD2 and SMAD3. We have corrected the figure caption of Figure 4 in the revised manuscript. (page 6, line 168)
Comments 8: Lines 153-154: The authors mention that phosphorylated SMAD2 and SMAD3 levels were elevated after GDNF treatment for 4 hours, but they do not show a vehicle control for the 4-hour treatment. This control is necessary to determine hether the increase in phosphorylated SMAD is a normal cellular response.
Response 8: Thanks for your valuable comments. Herein, the period of GDNF treatment was determined according to the previously published article (PMID: 31171625, title: Glial cell line-derived neurotrophic factor (GDNF) mediates hepatic stellate cell activation via ALK5/Smad signalling). This paper has been added to the reference in the revised manuscript. (page 6, line 157)
Even so, we agree with your perspective. It is true that an accurate control will make the data more elaborate and convincing. This suggestion provides significant guidance for our researches.
Comments 9: Figure 5. The HE staining and immunohistochemistry images do not clearly reflect the results described in the text. The authors should provide higher magnification images (at least 20X) to enhance clarity.
Response 9: Thanks for your valuable comments. We have changed them to the images with higher magnification. Therefore, a new figure 5 was provided. The previous images were provided as a supplementary figure 2.
-Figure 5 (page 7, line 83)
-Figure S2 (supplementary materials)
Figure S2. GDNF accelerates GBM growth in vivo. (A) HE staining of tumor tissues. (B) Immunohistochemistry was performed to detect the levels of ki-67, GFAP, MMP2 and MMP9. Bar = 50 µm. n = 5.
Comments 10: Line 313: The authors mention that "Stable C6 cell lines obtained by flow cytometer were used for the subsequent construction of the mouse model." However, they do not present the results of this sorting by FACS. These results should be included as a supplementary figure at least.
Response 10: Thanks for your good advice. Thank you for pointing this out. Results of sorting cells by flow cytometer are presented as a supplementary figure 4 in the revised manuscript. (supplementary materials)
Figure S4. C6 cells were sorted by FACS based on SERPINE1 knockdown. (A) Negative control group. (B) RNAi-1 group. (C) RNAi-2 group. (D) RNAi-3 group.
Comments 11: The reliance on overexpression and knockdown experiments may introduce artifacts. For instance, the high concentrations of GDNF used in vitro might not accurately reflect physiological conditions. It would be beneficial to rigorously examine the dose-response relationship and use additional controls to rule out off-target effects.
Response 11: Thanks for your valuable comments. We agree with you. This suggestion is very beneficial for following study design. Although numerous studies have shown that GDNF is highly expressed in GBM tissues, the true concentration of GDNF in GBM patients has been unknown. Our previous investigation found that the amount of GDNF secreted by C6 cells was 5-6 times that of astrocytes. The results are shown in the figure below.
Some studies used 10 ng/ml (PMID: 31171625) or 200 ng/ml GDNF (PMID: 35925441). Some studies even used 1000 ng/ml GDNF (PMID: 33404121). High-concentration GDNF may not accurately reflect physiological conditions. However, interestingly, we found that the role of GDNF in promoting GBM growth is very obvious in vivo studies, and the dose required is very low. These data indicate that GDNF plays an important role in GBM progression. To make the results more accurate and convincing, more attention will be paid to rigorous dose-response and control for in vitro researches, meanwhile more in vivo investigation will be carried out to verify in vitro findings.
Comments 12: The in vivo experiments use a subcutaneous xenograft model, which does not fully replicate the brain tumor microenvironment. The use of orthotopic GBM models would provide a more accurate assessment of tumor behavior and its interaction with surrounding brain tissues.
Response 12: Thanks for your good suggestions. We agree with you. In the present study, a mouse subcutaneous xenograft glioma model was established to assess the effect of GDNF on GBM growth. This model clearly determined the influence of GDNF and SERPINE1 on the volume and weight of GBM tumors (PMID: 33278485). To uncover the role of GDNF in GBM cell migration and invasion using this model, IHC assay was performed to detect the expression of migration protein markers such GFAP, MMP2 and MMP9. Indeed, the orthotopic GBM models can assess tumor behavior more accurately. We have pointed out this deficiency in the “Discussion” section. (page 8, line 232-236) In our following researches, we will utilize more animal models to explore the effects of GDNF on GBM progression. In fact, our another manuscript entitled “Hypoxia-driven GDNF-HIF-1α loop reinforces glycolysis in glioblastoma” will use the orthotopic GBM model to identify the roles of GDNF (unpublished).
Comments 13: A concluding paragraph should be added at the end of the discussion section to summarize the key findings and their implications.
Response 13: Thanks for your good proposal. We have added a concluding paragraph at the end of the “Discussion” section. (page 9, line 264-266) The details are as followed.
-In conclusion, our study illustrated that GDNF reinforced GBM cell migration and invasion through SMAD2/3-SERPINE1 axis. These results deepen the understanding of GDNF promoting GBM progression, and provide new ideas for GBM treatment.
Minor Revisions:
Comments 14: Figure 1D. It is unclear whether the plot shows overall survival or disease-free survival. This should be clearly stated in the figure caption and corrected accordingly.
Response 14: Thanks for your good advice. This plot shows the overall survival of GBM patients. We have stated it in the Figure 1 and its caption. Therefore, a new Figure 1 were provided. Additionally, we have corrected it in the section of “Results-2.1”. The details are as followed.
-Figure (page 3, line 96)
-Figure caption (page 3, line 100)
Overall survival was analyzed to evaluate the association between the mRNA level of SERPINE1 and the outcome of GBM patients by oncomine website.
-Results-2.1 (page 2, line 88)
The survival analysis indicated that overall survival of GBM patients was low in high SERPINE1 expression group compared with low SERPINE1 expression group (Figure 1D).
Comments 15: Figure 1E. The figure caption does not clarify what "NB-1," "NB-2," and "NB-3" represent. Please specify this.
Response 15: Thanks for your careful work. Sorry. It is an oversight in our manuscript. “NB” represents normal human brain. We have specified it in the section of “Results”, “Figure caption” (page 3, line 103) and “Materials-4.1” (page 9, line 282-285).
-Materials-4.1
GBM and NB tissues were acquired from Affiliated Hospital of Xuzhou Medical University. Three GBM tissues were collected from the surgical specimen archives, and three NB tissues were obtained from the patients who suffered acute brain damage and underwent intracranial decompression.
Comments 16: The authors should minimize the use of adjectives such as "strikingly" and "evidently" throughout the text.
Response 16: Thanks for your suggestion. We have minimized the use of these adjectives in the revised manuscript.
Comments 17: The manuscript should be carefully reviewed for typographical errors.
Response 17: Thank you very much for your comments. We have checked our manuscript carefully and improved it.
Comments 18: Abbreviations should be defined or written in full upon their first appearance only. For instance, "FBS" is first mentioned on line 260 and repeated in line 265.
Response 18: Thanks for your careful work. We have checked the abbreviations throughout the manuscript and revised it.

Reviewer 2 Report
Comments and Suggestions for Authors
In manuscript “GDNF promotes glioblastoma cell migration and invasion via SMAD2/3-SERPINE1 signaling axis”, the authors explored the promotion effect of GNDF on GBM progression, especially migration and invasion and uncovered the mechanism. They found that GDNF enhances GBM cell invasion and migration through SMAD2/3-SERPRINE1 pathway. This study is interesting and significant. However, it can be further improved with the following comments:
1) In figure 1E, are those normal tissue and GBM tissues paired samples which are from the same person?
2) For the western blot data, the GAPDH bands are over exposed. Please post a short time exposure one.
3) In figure 2, for U251 cells, it doesn’t show a good dose response. 5 and 10 ng/ml treatment may be needed.
4) In figure 3, for C6 cells, the western blot data don’t match with ELISA data. 80 ng/ml GDNF induced the highest response in western blot data, while 100 ng/ml for ELISA data.
5) In figure 3 and 4, cell migration protein markers should be characterized.
6) In figure 5, the authors tested the acceleration of GDNF on GBM growth using a xenograft mouse model. The invasion model could be better, given that this paper focused on migration and invasion.
Comments on the Quality of English LanguageIt can be improved.
Author Response
Comments and Suggestions for Authors
In manuscript “GDNF promotes glioblastoma cell migration and invasion via SMAD2/3-SERPINE1 signaling axis”, the authors explored the promotion effect of GNDF on GBM progression, especially migration and invasion and uncovered the mechanism. They found that GDNF enhances GBM cell invasion and migration through SMAD2/3-SERPRINE1 pathway. This study is interesting and significant. However, it can be further improved with the following comments:
- Summary
Thank you very much for your valuable comments concerning our manuscript entitled “GDNF promotes glioblastoma cell migration and invasion via SMAD2/3-SERPINE1 signaling axis” (ID: ijms-3169672). These comments help us to improve our manuscript, and provide significant guidance for our study. We have carefully studied these comments and made corrections. Revised portion are marked in red in the manuscript. The main corrections in the manuscript and responses to the comments are as followed.
2. Questions for General Evaluation |
Reviewer’s Evaluation |
Response and Revisions |
Does the introduction provide sufficient background and include all relevant references? |
Yes |
|
Is the research design appropriate? |
Can be improved |
Agree. We have revised it. |
Are the methods adequately described? |
Yes |
|
Are the results clearly presented? |
Yes |
|
Are the conclusions supported by the results? |
Can be improved |
Agree. We have revised it. |
- Point-by-point response to Comments and Suggestions for Authors
Comments 1: In figure 1E, are those normal tissue and GBM tissues paired samples which are from the same person?
Response 1: Thank you very much for your comments. In figure 1E, normal brain (NB) tissues and GBM tissues are not paired samples from the same person. Sorry. It is an oversight in our manuscript. We have clearly stated it in the section of “Materials and methods-4.1” (page 9, line 282-285). The details are as followed.
-GBM and NB tissues were acquired from Affiliated Hospital of Xuzhou Medical University. Three GBM tissues were collected from the surgical specimen archives, and three NB tissues were obtained from the patients who suffered acute brain damage and underwent intracranial decompression.
Comments 2: For the western blot data, the GAPDH bands are over exposed. Please post a short time exposure one.
Response 2: Thanks for your valuable suggestions. The GAPDH band has been changed to a short time exposure one. Therefore, a new Figure 2 were provided. (page 4, line 118)
Comments 3: In figure 2, for U251 cells, it doesn’t show a good dose response. 5 and 10 ng/ml treatment may be needed.
Response 3: Thanks for your valuable comments. We agree with your perspective. The data will be more elaborate if more GDNF concentrations were used. In the present study, the concentrations of GDNF were selected based on our published article (PMID: 29088765, title: Neuropilin-1 is a glial cell line-derived neurotrophic factor receptor in glioblastoma). This paper has been added to the reference in the revised manuscript (page 3, line 109). In this paper, C6 cells were treated with various concentrations of GDNF (0, 0.1, 1, 10, 20, 40, 80, 100, 200 and 400 ng/ml) and grown for 24, 48 or 72h. The detailed results are shown in the figure below. To determine the most optimal concentration of GDNF to promote SERPINE1 expression in the present work, the protein and secretion levels of SERPINE1 were separately examined by western blotting and ELISA, after cells were treated with various GDNF concentrations (0, 20, 40, 80, 100 ng/ml) for 48 h.
Comments 4: In figure 2, for C6 cells, the western blot data don’t match with ELISA data. 80 ng/ml GDNF induced the highest response in western blot data, while 100 ng/ml for ELISA data.
Response 4: Thank you for your valuable comments. We also noticed this phenomenon. The possible reason is that SERPINE1 expressed inside the cells and secreted outside the cells peak at different time. In this study, the results of western blotting and ELISA were combined to determine the most optimal GDNF concentration.
Comments 5: In figure 3 and 4, cell migration protein markers should be characterized.
Response 5: Thanks for your good suggestions. We agree with you. Detection of proteins associated with cell migration and invasion is a good complement to the results of in vitro functional experiments. In vivo experiments, we used IHC staining to detect the expression of migration protein markers such GFAP, MMP2 and MMP9. Therefore, this study indicated that GDNF promoted GBM cell migration and invasion by SERPINE1 according to the results of in vitro and in vivo experiments. Additionally, this suggestion is valuable for our future researches. At present, there are two ongoing studies on GDNF promoting malignant progression of GBM. We will add the detection of protein markers to enrich our results.
Comments 6: In figure 5, the authors tested the acceleration of GDNF on GBM growth using a xenograft mouse model. The invasion model could be better, given that this paper focused on migration and invasion.
Response 6: Thanks for your good proposal. We agree with you. In the present study, a mouse subcutaneous xenograft glioma model was established to assess the effect of GDNF on GBM growth. This model clearly determined the influence of GDNF and SERPINE1 on the volume and weight of GBM tumors (PMID: 33278485, title: Enhancer II-targeted dsRNA decreases GDNF expression via histone H3K9 trimethylation to inhibit glioblastoma progression). To uncover the roles of GDNF in GBM cell migration and invasion using this model, IHC assay was performed to detect the expression of migration protein markers such GFAP, MMP2 and MMP9. Indeed, the invasion model is better for studying cell migration and invasion. We have pointed out this deficiency in the “Discussion” section (page 8, line 232-236). In our following researches, we will utilize more animal models to explore the effects of GDNF on GBM progression. In fact, our another manuscript entitled “Hypoxia-driven GDNF-HIF-1α loop reinforces glycolysis in glioblastoma” will use the orthotopic GBM model to identify the roles of GDNF (unpublished).
The newly added discussion is as followed.
-To verify the results of in vitro experiments, mouse subcutaneous xenograft glioma model was established to assess the effects of GDNF and SERPINE1 on GBM growth in this study. However, the orthotopic GBM models could assess tumor behaviors more accurately. In the future researches, we will utilize more animal models to explore the effects of GDNF on GBM progression.
- Response to Comments on the Quality of English Language
Point 1: It can be improved.
Response 1: Agree. Although we have tried our best to improve the English Language, this manuscript still requires improvement to the language. We accept MDPI Author Services.
Once again, special thanks to you for your valuable comments.

Round 2
Reviewer 2 Report
Comments and Suggestions for Authors
It gets improved.